# Inverse Reinforcement Learning with Locally Consistent Reward Functions

**Quoc Phong Nguyen[†], Kian Hsiang Low[†], and Patrick Jaillet[§]**
Dept. of Computer Science, National University of Singapore, Republic of Singapore[†]
Dept. of Electrical Engineering and Computer Science, Massachusetts Institute of Technology, USA[§]
{qphong,lowkh}@comp.nus.edu.sg[†], jaillet@mit.edu[§]

## Abstract

Existing *inverse reinforcement learning* (IRL) algorithms have assumed each expert's demonstrated trajectory to be produced by only a single reward function. This paper presents a novel generalization of the IRL problem that allows each trajectory to be generated by multiple locally consistent reward functions, hence catering to more realistic and complex experts' behaviors. Solving our generalized IRL problem thus involves not only learning these reward functions but also the stochastic transitions between them at any state (including unvisited states). By representing our IRL problem with a probabilistic graphical model, an *expectation-maximization* (EM) algorithm can be devised to iteratively learn the different reward functions and the stochastic transitions between them in order to jointly improve the likelihood of the expert's demonstrated trajectories. As a result, the most likely partition of a trajectory into segments that are generated from different locally consistent reward functions selected by EM can be derived. Empirical evaluation on synthetic and real-world datasets shows that our IRL algorithm outperforms the state-of-the-art EM clustering with maximum likelihood IRL, which is, interestingly, a reduced variant of our approach.

## 1   Introduction

The reinforcement learning problem in *Markov decision processes* (MDPs) involves an agent using its observed rewards to learn an optimal policy that maximizes its expected total reward for a given task. However, such observed rewards or the reward function defining them are often not available nor known in many real-world tasks. The agent can therefore learn its reward function from an expert associated with the given task by observing the expert's behavior or demonstration, and this approach constitutes the *inverse reinforcement learning* (IRL) problem.

Unfortunately, the IRL problem is ill-posed because infinitely many reward functions are consistent with the expert's observed behavior. To resolve this issue, existing IRL algorithms have proposed alternative choices of the agent's reward function that minimize different dissimilarity measures defined using various forms of abstractions of the agent's generated optimal behavior vs. the expert's observed behavior, as briefly discussed below (see [17] for a detailed review): (a) The projection algorithm [1] selects a reward function that minimizes the squared Euclidean distance between the feature expectations obtained by following the agent's generated optimal policy and the empirical feature expectations observed from the expert's demonstrated state-action trajectories; (b) the multiplicative weights algorithm for apprentice learning [24] adopts a robust minimax approach to deriving the agent's behavior, which is guaranteed to perform no worse than the expert and is equivalent to choosing a reward function that minimizes the difference between the expected average reward under the agent's generated optimal policy and the expert's empirical average reward approximated using the agent's reward weights; (c) the linear programming apprentice learning algorithm [23] picks its reward function by minimizing the same dissimilarity measure but incurs much less time empirically; (d) the policy matching algorithm [16] aims to match the agent's generated optimal behavior to the expert's observed behavior by choosing a reward function that minimizes the sum of

squared Euclidean distances between the agent's generated optimal policy and the expert's estimated policy (i.e., from its demonstrated trajectories) over every possible state weighted by its empirical state visitation frequency; (e) the maximum entropy IRL [27] and *maximum likelihood IRL* (MLIRL) [2] algorithms select reward functions that minimize an empirical approximation of the Kullback-Leibler divergence between the distributions of the agent's and expert's generated state-action trajectories, which is equivalent to maximizing the average log-likelihood of the expert's demonstrated trajectories. The log-likelihood formulations of the maximum entropy IRL and MLIRL algorithms differ in the use of smoothing at the trajectory and action levels, respectively. As a result, the former's log-likelihood or dissimilarity measure does not utilize the agent's generated optimal policy, which is consequently questioned by [17] as to whether it is considered an IRL algorithm. Bayesian IRL [21] extends IRL to the Bayesian setting by maintaining a distribution over all possible reward functions and updating it using Bayes rule given the expert's demonstrated trajectories. The work of [5] extends the projection algorithm [1] to handle partially observable environments given the expert's policy (i.e., represented as a finite state controller) or observation-action trajectories.

All the IRL algorithms described above have assumed that the expert's demonstrated trajectories are only generated by a single reward function. To relax this restrictive assumption, the recent works of [2, 6] have, respectively, generalized MLIRL (combining it with *expectation-maximization* (EM) clustering) and Bayesian IRL (integrating it with a Dirichlet process mixture model) to handle trajectories generated by multiple reward functions (e.g., due to many intentions) in observable environments. But, each trajectory is assumed to be produced by a single reward function.

In this paper, we propose a new generalization of the IRL problem in observable environments, which is inspired by an open question posed in the seminal works of IRL [19, 22]: If behavior is strongly inconsistent with optimality, can we identify "locally consistent" reward functions for specific regions in state space? Such a question implies that no single reward function is globally consistent with the expert's behavior, hence invalidating the use of all the above-mentioned IRL algorithms. More importantly, multiple reward functions may be locally consistent with the expert's behavior in different segments along its state-action trajectory and the expert has to switch/transition between these locally consistent reward functions during its demonstration. This can be observed in the following real-world example [26] where every possible intention of the expert is uniquely represented by a different reward function: A driver intends to take the highway to a food center for lunch. An electronic toll coming into effect on the highway may change his intention to switch to another route. Learning of the driver's intentions to use different routes and his transitions between them allows the transport authority to analyze, understand, and predict the traffic route patterns and behavior for regulating the toll collection. This example, among others (e.g., commuters' intentions to use different transport modes, tourists' intentions to visit different attractions, Section 4), motivate the practical need to formalize and solve our proposed generalized IRL problem.

This paper presents a novel generalization of the IRL problem that, in particular, allows each expert's state-action trajectory to be generated by multiple locally consistent reward functions, hence catering to more realistic and complex experts' behaviors than that afforded by existing variants of the IRL problem (which all assume that each trajectory is produced by a single reward function) discussed earlier. At first glance, one may straightaway perceive our generalization as an IRL problem in a partially observable environment by representing the choice of locally consistent reward function in a segment as a latent state component. However, the observation model cannot be easily specified nor learned from the expert's state-action trajectories, which invalidates the use of IRL for POMDP [5]. Instead, we develop a probabilistic graphical model for representing our generalized IRL problem (Section 2), from which an EM algorithm can be devised to iteratively select the locally consistent reward functions as well as learn the stochastic transitions between them in order to jointly improve the likelihood of the expert's demonstrated trajectories (Section 3). As a result, the most likely partition of an expert's demonstrated trajectory into segments that are generated from different locally consistent reward functions selected by EM can be derived (Section 3), thus enabling practitioners to identify states in which the expert transitions between locally consistent reward functions and investigate the resulting causes. To extend such a partitioning to work for trajectories traversing through any (possibly unvisited) region of the state space, we propose using a generalized linear model to represent and predict the stochastic transitions between reward functions at any state (i.e., including states not visited in the expert's demonstrated trajectories) by exploiting features that influence these transitions (Section 2). Finally, our proposed IRL algorithm is empirically evaluated using both synthetic and real-world datasets (Section 4).

## 2 Problem Formulation

A *Markov decision process* (MDP) for an agent is defined as a tuple $(\mathcal{S}, \mathcal{A}, t, r_\theta, \gamma)$ consisting of a finite set $\mathcal{S}$ of its possible states such that each state $s \in \mathcal{S}$ is associated with a column vector $\phi_s$ of realized feature measurements, a finite set $\mathcal{A}$ of its possible actions, a state transition function $t : \mathcal{S} \times \mathcal{A} \times \mathcal{S} \to [0, 1]$ denoting the probability $t(s, a, s') \triangleq P(s'|s, a)$ of moving to state $s'$ by performing action $a$ in state $s$, a reward function $r_\theta : \mathcal{S} \to \mathbb{R}$ mapping each state $s \in \mathcal{S}$ to its reward $r_\theta(s) \triangleq \theta^\top \phi_s$ where $\theta$ is a column vector of reward weights, and constant factor $\gamma \in (0, 1)$ discounting its future rewards. When $\theta$ is known, the agent can compute its policy $\pi_\theta : \mathcal{S} \times \mathcal{A} \to [0, 1]$ specifying the probability $\pi_\theta(s, a) \triangleq P(a|s, r_\theta)$ of performing action $a$ in state $s$. However, $\theta$ is not known in IRL and to be learned from an expert (Section 3).

Let $\mathcal{R}$ denote a finite set of locally consistent reward functions of the agent and $r_{\widetilde{\theta}}$ be a reward function chosen arbitrarily from $\mathcal{R}$ prior to learning. Define a transition function $\tau_\omega : \mathcal{R} \times \mathcal{S} \times \mathcal{R} \to [0, 1]$ for switching between these reward functions as the probability $\tau_\omega(r_\theta, s, r_{\theta'}) \triangleq P(r_{\theta'}|s, r_\theta, \omega)$ of switching from reward function $r_\theta$ to reward function $r_{\theta'}$ in state $s$ where the set $\omega \triangleq \{\omega_{r_\theta r_{\theta'}}\}_{r_\theta \in \mathcal{R}, r_{\theta'} \in \mathcal{R} \setminus \{r_{\widetilde{\theta}}\}}$ contains column vectors of transition weights $\omega_{r_\theta r_{\theta'}}$ for all $r_\theta \in \mathcal{R}$ and $r_{\theta'} \in \mathcal{R} \setminus \{r_{\widetilde{\theta}}\}$ if the features influencing the stochastic transitions between reward functions can be additionally observed by the agent during the expert's demonstration, and $\omega \triangleq \emptyset$ otherwise. In our generalized IRL problem, $\tau_\omega$ is not known and to be learned from the expert (Section 3). Specifically, in the former case, we propose using a generalized linear model to represent $\tau_\omega$:

$$\tau_\omega(r_\theta, s, r_{\theta'}) \triangleq \begin{cases} \exp(\omega_{r_\theta r_{\theta'}}^\top \varphi_s)/(1 + \sum_{r_{\bar{\theta}} \in \mathcal{R} \setminus \{r_{\widetilde{\theta}}\}} \exp(\omega_{r_\theta r_{\bar{\theta}}}^\top \varphi_s)) & \text{if } r_{\theta'} \neq r_{\widetilde{\theta}}, \\ 1/(1 + \sum_{r_{\bar{\theta}} \in \mathcal{R} \setminus \{r_{\widetilde{\theta}}\}} \exp(\omega_{r_\theta r_{\bar{\theta}}}^\top \varphi_s)) & \text{otherwise}; \end{cases} \tag{1}$$

where $\varphi_s$ is a column vector of random feature measurements influencing the stochastic transitions between reward functions (i.e., $\tau_\omega$) in state $s$.

*Remark* 1. Different from $\phi_s$ whose feature measurements are typically assumed in IRL algorithms to be realized/known to the agent for all $s \in \mathcal{S}$ and remain static over time, the feature measurements of $\varphi_s$ are, in practice, often not known to the agent *a priori* and can only be observed when the expert (agent) visits the corresponding state $s \in \mathcal{S}$ during its demonstration (execution), and may vary over time according to some unknown distribution, as motivated by the real-world examples given in Section 1. Without prior observation of the feature measurements of $\varphi_s$ for all $s \in \mathcal{S}$ (or knowledge of their distributions) necessary for computing $\tau_\omega$ (1), the agent cannot consider exploiting $\tau_\omega$ for switching between reward functions within MDP or POMDP planning, even after learning its weights $\omega$; this eliminates the possibility of reducing our generalized IRL problem to an equivalent conventional IRL problem (Section 1) with only a single reward function (i.e., comprising a mixture of locally consistent reward functions). Furthermore, the observation model cannot be easily specified nor learned from the expert's trajectories of states, actions, and $\varphi_s$, which invalidates the use of IRL for POMDP [5]. Instead of exploiting $\tau_\omega$ within planning, during the agent's execution, when it visits some state $s$ and observes the feature measurements of $\varphi_s$, it can then use and compute $\tau_\omega$ for state $s$ to switch between reward functions, each of which has generated a separate MDP policy prior to execution, as illustrated in a simple example in Fig. 1 below.

*Remark* 2. Using a generalized linear model to represent $\tau_\omega$ (1) allows learning of the stochastic transitions between reward functions (specifically, by learning $\omega$ (Section 3)) to be generalized across different states. After learning, (1) can then be exploited for predicting the stochastic transitions between reward functions at any state (i.e., including states not visited in the expert's demonstrated state-action trajectories). Consequently, the agent can choose to traverse a trajectory through any region (i.e., possibly not visited by the expert) of the state space during its execution and the most likely partition of its trajectory into segments that are generated from different locally consistent reward functions selected by EM can still be derived (Section 3). In contrast, if the feature measurements of $\varphi_s$ cannot be observed by the agent during the expert's demonstration (i.e., $\omega = \emptyset$, as defined above), then such a generalization is not possible; only the transition probabilities of switching between reward functions at states visited in the expert's demonstrated trajectories can be estimated (Section 3). In practice, since the number $|\overline{\mathcal{S}}|$ of visited states is expected to be much larger than the length $L$ of any feature vector $\varphi_s$,

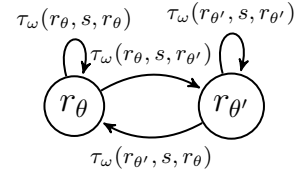

Figure 1: Transition function $\tau_\omega$ of an agent in state $s$ for switching between two reward functions $r_\theta$ and $r_{\theta'}$ with their respective policies $\pi_\theta$ and $\pi_{\theta'}$ generated prior to execution.

the number $\mathcal{O}(|\overline{\mathcal{S}}||\mathcal{R}|^2)$ of transition probabilities to be estimated is bigger than $|\omega| = \mathcal{O}(L|\mathcal{R}|^2)$ in (1). So, observing $\varphi_s$ offers a further advantage of reducing the number of parameters to be learned.

Fig. 2 shows the probabilistic graphical model for representing our generalized IRL problem. To describe our model, some notations are necessary: Let $N$ be the number of the expert's demonstrated trajectories and $T_n$ be the length (i.e., number of time steps) of its $n$-th trajectory for $n = 1, \ldots, N$. Let $r_{\theta_t^n} \in \mathcal{R}$, $a_t^n \in \mathcal{A}$, and $s_t^n \in \mathcal{S}$ denote its reward function, action, and state at time step $t$ in its $n$-th trajectory, respectively. Let $R_{\theta_t^n}$, $A_t^n$, and $S_t^n$ be random variables corresponding to their respective realizations $r_{\theta_t^n}$, $a_t^n$, and $s_t^n$ where $R_{\theta_t^n}$ is a latent variable, and $A_t^n$ and $S_t^n$ are observable variables. Define $r_{\theta^n} \triangleq (r_{\theta_t^n})_{t=0}^{T_n}$, $a^n \triangleq (a_t^n)_{t=1}^{T_n}$, and $s^n \triangleq (s_t^n)_{t=1}^{T_n}$ as sequences of all its reward functions, actions, and states in its $n$-th trajectory, respectively. Finally, define $r_{\theta^{1:N}} \triangleq (r_{\theta^n})_{n=1}^{N}$, $a^{1:N} \triangleq (a^n)_{n=1}^{N}$, and $s^{1:N} \triangleq (s^n)_{n=1}^{N}$ as tuples of all its reward function sequences, action sequences, and state sequences in its $N$ trajectories, respectively.

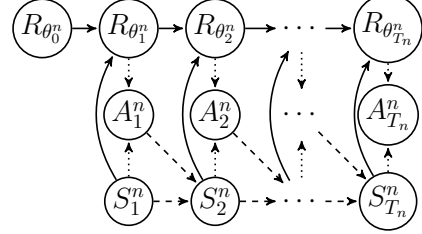

Figure 2: Probabilistic graphical model of the expert's $n$-th demonstrated trajectory encoding its stochastic transitions between reward functions with *solid edges* (i.e., $\tau_\omega(r_{\theta_{t-1}^n}, s_t^n, r_{\theta_t^n}) = P(r_{\theta_t^n}|s_t^n, r_{\theta_{t-1}^n}, \omega)$ for $t = 1, \ldots, T_n$), state transitions with *dashed edges* (i.e., $t(s_t^n, a_t^n, s_{t+1}^n) = P(s_{t+1}^n|s_t^n, a_t^n)$ for $t = 1, \ldots, T_n - 1$), and policy with *dotted edges* (i.e., $\pi_{\theta_t^n}(s_t^n, a_t^n) = P(a_t^n|s_t^n, r_{\theta_t^n})$ for $t = 1, \ldots, T_n$).

It can be observed from Fig. 2 that our probabilistic graphical model of the expert's $n$-th demonstrated trajectory encodes its stochastic transitions between reward functions, state transitions, and policy. Through our model, the Viterbi algorithm [20] can be applied to derive the most likely partition of the expert's trajectory into segments that are generated from different locally consistent reward functions selected by EM, as shown in Section 3. Given the state transition function $t(\cdot, \cdot, \cdot)$ and the number $|\mathcal{R}|$ of reward functions, our model allows tractable learning of the unknown parameters using EM (Section 3), which include the reward weights vector $\theta$ for all reward functions $r_\theta \in \mathcal{R}$, transition function $\tau_\omega$ for switching between reward functions, initial state probabilities $\nu(s) \triangleq P(S_1^n = s)$ for all $s \in \mathcal{S}$, and initial reward function probabilities $\sigma(r_\theta) \triangleq P(R_{\theta_0^n} = r_\theta)$ for all $r_\theta \in \mathcal{R}$.

## 3 EM Algorithm for Parameter Learning

A straightforward approach to learning the unknown parameters $\Lambda \triangleq (\nu, \sigma, \{\theta|r_\theta \in \mathcal{R}\}, \tau_\omega)$ is to select the value of $\Lambda$ that directly maximizes the log-likelihood of the expert's demonstrated trajectories. Computationally, such an approach is prohibitively expensive due to a large joint parameter space to be searched for the optimal value of $\Lambda$. To ease this computational burden, our key idea is to devise an EM algorithm that iteratively refines the estimate for $\Lambda$ to improve the *expected* log-likelihood instead, which is guaranteed to improve the original log-likelihood by at least as much:

**Expectation (E) step.** $Q(\Lambda, \Lambda^i) \triangleq \sum_{r_{\theta^{1:N}}} P(r_{\theta^{1:N}}|s^{1:N}, a^{1:N}, \Lambda^i) \log P(r_{\theta^{1:N}}, s^{1:N}, a^{1:N}|\Lambda)$.

**Maximization (M) step.** $\Lambda^{i+1} = \text{argmax}_\Lambda Q(\Lambda, \Lambda^i)$

where $\Lambda^i$ denotes an estimate for $\Lambda$ at iteration $i$. The $Q$ function of EM can be reduced to the following sum of five terms, as shown in Appendix A:

$$Q(\Lambda, \Lambda^i) = \sum_{n=1}^{N} \log \nu(s_1^n) + \sum_{n=1}^{N} \sum_{r_\theta \in \mathcal{R}} P(R_{\theta_0^n} = r_\theta|s^n, a^n, \Lambda^i) \log \sigma(r_\theta) \tag{2}$$

$$+ \sum_{n=1}^{N} \sum_{t=1}^{T_n} \sum_{r_\theta \in \mathcal{R}} P(R_{\theta_t^n} = r_\theta|s^n, a^n, \Lambda^i) \log \pi_{\theta}(s_t^n, a_t^n) \tag{3}$$

$$+ \sum_{n=1}^{N} \sum_{t=1}^{T_n} \sum_{r_\theta, r_{\theta'} \in \mathcal{R}} P(R_{\theta_{t-1}^n} = r_\theta, s_t^n, R_{\theta_t^n} = r_{\theta'}|s^n, a^n, \Lambda^i) \times \log \tau_\omega(r_\theta, s_t^n, r_{\theta'}) \tag{4}$$

$$+ \sum_{n=1}^{N} \sum_{t=1}^{T_n - 1} \log t(s_t^n, a_t^n, s_{t+1}^n) . \tag{5}$$

Interestingly, each of the first four terms in (2), (3), and (4) contains a unique unknown parameter type (respectively, $\nu$, $\sigma$, $\{\theta|r_\theta \in \mathcal{R}\}$, and $\tau_\omega$) and can therefore be maximized separately in the M step to be discussed below. As a result, the parameter space to be searched can be greatly reduced. Note that the third term (3) generalizes the log-likelihood in MLIRL [2] (i.e., assuming all trajectories to be produced by a single reward function) to that allowing each expert's trajectory to be generated by multiple locally consistent reward functions. The last term (5), which contains the known state transition function $t$, is independent of unknown parameters $\Lambda$.[1]

**Learning initial state probabilities.** To maximize the first term in the $Q$ function (2) of EM, we use the method of Lagrange multipliers with the constraint $\sum_{s \in \mathcal{S}} \nu(s) = 1$ to obtain the estimate $\widehat{\nu}(s) = (1/N) \sum_{n=1}^{N} I_1^n$ for all $s \in \mathcal{S}$ where $I_1^n$ is an indicator variable of value 1 if $s_1^n = s$, and 0 otherwise. Since $\widehat{\nu}$ can be computed directly from the expert's demonstrated trajectories in $\mathcal{O}(N)$ time, it does not have to be refined.

**Learning initial reward function probabilities.** To maximize the second term in $Q$ function (2) of EM, we utilize the method of Lagrange multipliers with the constraint $\sum_{r_\theta \in \mathcal{R}} \sigma(r_\theta) = 1$ to derive

$$\sigma^{i+1}(r_{\theta^i}) = (1/N) \sum_{n=1}^{N} P(R_{\theta_0^n} = r_\theta | s^n, a^n, \Lambda^i) \tag{6}$$

for all $r_{\theta^i} \in \mathcal{R}$ where $\sigma^{i+1}$ denotes an estimate for $\sigma$ at iteration $i+1$, $\theta^i$ denotes an estimate for $\theta$ at iteration $i$, and $P(R_{\theta_t^n} = r_\theta | s^n, a^n, \Lambda^i)$ (in this case, $t = 0$) can be computed in $\mathcal{O}(\sum_{n=1}^{N} |\mathcal{R}|^2 T_n)$ time using a procedure inspired by Baum-Welch algorithm [3], as shown in Appendix B.

**Learning reward functions.** The third term in the $Q$ function (3) of EM is maximized using gradient ascent and its gradient $g_1(\theta)$ with respect to $\theta$ is derived to be

$$g_1(\theta) \triangleq \sum_{n=1}^{N} \sum_{t=1}^{T_n} \frac{P(R_{\theta_t^n} = r_\theta | s^n, a^n, \Lambda^i)}{\pi_\theta(s_t^n, a_t^n)} \frac{\mathrm{d}\pi_\theta(s_t^n, a_t^n)}{\mathrm{d}\theta} \tag{7}$$

for all $\theta \in \{\theta' | r_{\theta'} \in \mathcal{R}\}$. For $\pi_\theta(s_t^n, a_t^n)$ to be differentiable in $\theta$, we define the $Q_\theta$ function of MDP using an operator that blends the $Q_\theta$ values via Boltzmann exploration [2]: $Q_\theta(s, a) \triangleq \theta^\top \phi_s + \gamma \sum_{s' \in \mathcal{S}} t(s, a, s') \otimes_{a'} Q_\theta(s', a')$ where $\otimes_a Q_\theta(s, a) \triangleq \sum_{a \in \mathcal{A}} Q_\theta(s, a) \times \pi_\theta(s, a)$ such that $\pi_\theta(s, a) \triangleq \exp(\beta Q_\theta(s, a)) / \sum_{a' \in \mathcal{A}} \exp(\beta Q_\theta(s, a'))$ is defined as a Boltzmann exploration policy, and $\beta > 0$ is a temperature parameter. Then, we update $\theta^{i+1} \leftarrow \theta^i + \delta g_1(\theta^i)$ where $\delta$ is the learning step size. We use backtracking line search method to improve the performance of gradient ascent. Similar to MLIRL, the time incurred in each iteration of gradient ascent depends mostly on that of value iteration, which increases with the size of the MDP's state and action space.

**Learning transition function for switching between reward functions.** To maximize the fourth term in the $Q$ function (4) of EM, if the feature measurements of $\varphi_s$ cannot be observed by the agent during the expert's demonstration (i.e., $\omega = \emptyset$), then we utilize the method of Lagrange multipliers with the constraints $\sum_{r_{\theta'} \in \mathcal{R}} \tau_\omega(r_\theta, s, r_{\theta'}) = 1$ for all $r_\theta \in \mathcal{R}$ and $s \in \overline{\mathcal{S}}$ to obtain

$$\tau_{\omega^{i+1}}(r_{\theta^i}, s, r_{\theta'^i}) = \left(\sum_{n=1}^{N} \sum_{t=1}^{T_n} \gamma_{n,t,r_{\theta^i},s,r_{\theta'^i}}\right) / \left(\sum_{r_{\bar{\theta}^i} \in \mathcal{R}} \sum_{n=1}^{N} \sum_{t=1}^{T_n} \gamma_{n,t,r_{\theta^i},s,r_{\bar{\theta}^i}}\right) \tag{8}$$

for $r_{\theta^i}, r_{\theta'^i} \in \mathcal{R}$ and $s \in \overline{\mathcal{S}}$ where $\overline{\mathcal{S}}$ is the set of states visited by the expert, $\tau_{\omega^{i+1}}$ is an estimate for $\tau_\omega$ at iteration $i + 1$, and $\gamma_{n,t,r_{\theta^i},s,r_{\bar{\theta}^i}} \triangleq P(R_{\theta_{t-1}^n} = r_\theta, S_t^n = s, R_{\theta_t^n} = r_{\bar{\theta}} | s^n, a^n, \Lambda^i)$ can be computed efficiently by exploiting the intermediate results from evaluating $P(R_{\theta_t^n} = r_\theta | s^n, a^n, \Lambda^i)$ described previously, as detailed in Appendix B.

On the other hand, if the feature measurements of $\varphi_s$ can be observed by the agent during the expert's demonstration, then recall that we use a generalized linear model to represent $\tau_\omega$ (1) (Section 2) and $\omega$ is the unknown parameter to be estimated. Similar to learning the reward weights vector $\theta$ for reward function $r_\theta$, we maximize the fourth term (4) in the $Q$ function of EM by using gradient ascent and its gradient $g_2(\omega_{r_\theta r_{\theta'}})$ with respect to $\omega_{r_\theta r_{\theta'}}$ is derived to be

$$g_2(\omega_{r_\theta r_{\theta'}}) \triangleq \sum_{n=1}^{N} \sum_{t=1}^{T_n} \sum_{r_{\bar{\theta}} \in \mathcal{R}} \frac{\gamma_{n,t,r_\theta,s_t^n,r_{\bar{\theta}}}}{\tau_\omega(r_\theta, s_t^n, r_{\bar{\theta}})} \frac{\mathrm{d}\tau_\omega(r_\theta, s_t^n, r_{\bar{\theta}})}{\mathrm{d}\omega_{r_\theta r_{\theta'}}} \tag{9}$$

for all $\omega_{r_\theta r_{\theta'}} \in \omega$. Let $\omega_{r_\theta r_{\theta'}}^i$ denote an estimate for $\omega_{r_\theta r_{\theta'}}$ at iteration $i$. Then, it is updated using $\omega_{r_\theta r_{\theta'}}^{i+1} \leftarrow \omega_{r_\theta r_{\theta'}}^i + \delta g_2(\omega_{r_\theta r_{\theta'}}^i)$ where $\delta$ is the learning step size. Backtracking line search method is also used to improve the performance of gradient ascent here. In both cases, the time incurred in each iteration $i$ is proportional to the number of $\gamma_{n,t,r_{\theta^i},s,r_{\bar{\theta}^i}}$ to be computed, which is $\mathcal{O}(\sum_{n=1}^{N} |R|^2 |S| T_n)$ time.

**Viterbi algorithm for partitioning a trajectory into segments with different locally consistent reward functions.** Given the final estimate $\widehat{\Lambda} = (\widehat{\nu}, \widehat{\sigma}, \{\widehat{\theta} | r_{\widehat{\theta}} \in \mathcal{R}\}, \tau_{\widehat{\omega}})$ for the unknown parameters $\Lambda$ produced by EM, the most likely partition of the expert's $n$-th demonstrated trajectory into segments generated by different locally consistent reward functions is $r_{\theta^n}^* = (r_{\theta_t^n}^*)_{t=0}^{T_n} \triangleq \arg\max_{r_{\theta^n}} P(r_{\theta^n} | s^n, a^n, \widehat{\Lambda}) = \arg\max_{r_{\theta^n}} P(r_{\theta^n}, s^n, a^n | \widehat{\Lambda})$, which can be derived using the Viterbi algorithm [20]. Specifically, define $v_{r_{\bar{\theta}}, T}$ for $T = 1, \ldots, T_n$ as the probability of the most

likely reward function sequence $(r_{\theta_t^n})_{t=0}^{T-1}$ from time steps $0$ to $T-1$ ending with reward function $r_{\widehat{\theta}}$ at time step $T$ that produce state and action sequences $(s_t^n)_{t=1}^T$ and $(a_t^n)_{t=1}^T$:

$$v_{r_{\widehat{\theta}},T} \triangleq \max_{(r_{\theta_t^n})_{t=0}^{T-1}} P((r_{\theta_t^n})_{t=0}^{T-1}, R_{\theta_T^n} = r_\theta, (s_t^n)_{t=1}^T, (a_t^n)_{t=1}^T | \widehat{\Lambda})$$
$$= t(s_{T-1}^n, a_{T-1}^n, s_T^n)\, \pi_{\widehat{\theta}}(s_T^n, a_T^n) \max_{r_{\widehat{\theta}'}}\, v_{r_{\widehat{\theta}'},T-1}\, \tau_{\widehat{\omega}}(r_{\widehat{\theta}'}, s_T^n, r_{\widehat{\theta}}) \,,$$

$$v_{r_{\widehat{\theta}},1} \triangleq \max_{r_{\theta_0^n}} P(r_{\theta_0^n}, R_{\theta_1^n} = r_\theta, s_1^n, a_1^n | \widehat{\Lambda}) = \widehat{\nu}(s_1^n)\, \pi_{\widehat{\theta}}(s_1^n, a_1^n) \max_{r_{\widehat{\theta}'}}\, \widehat{\sigma}(r_{\widehat{\theta}'})\, \tau_{\widehat{\omega}}(r_{\widehat{\theta}'}, s_1^n, r_{\widehat{\theta}}) \,.$$

Then, $r_{\theta_0^n}^* = \operatorname{argmax}_{r_{\widehat{\theta}'}} \widehat{\sigma}(r_{\widehat{\theta}'})\, \tau_{\widehat{\omega}}(r_{\widehat{\theta}'}, s_1^n, r_{\theta_1^n}^*)$, $r_{\theta_T^n}^* = \operatorname{argmax}_{r_{\widehat{\theta}'}} v_{r_{\widehat{\theta}'},T}\, \tau_{\widehat{\omega}}(r_{\widehat{\theta}'}, s_{T+1}^n, r_{\theta_{T+1}^n}^*)$ for $T = 1, \ldots, T_n - 1$, and $r_{\theta_{T_n}^n}^* = \operatorname{argmax}_{r_{\widehat{\theta}}} v_{r_{\widehat{\theta}},T_n}$. The above Viterbi algorithm can be applied in the same way to partition an agent's trajectory traversing through any region (i.e., possibly not visited by the expert) of the state space during its execution in $\mathcal{O}(|\mathcal{R}|^2 T)$ time.

# 4 Experiments and Discussion

This section evaluates the empirical performance of our IRL algorithm using 3 datasets featuring experts' demonstrated trajectories in two simulated grid worlds and real-world taxi trajectories. The average log-likelihood of the expert's demonstrated trajectories is used as the performance metric because it inherently accounts for the fidelity of our IRL algorithm in learning the locally consistent reward functions (i.e., $\mathcal{R}$) and the stochastic transitions between them (i.e., $\tau_\omega$):

$$L(\Lambda) \triangleq (1/N_{\text{tot}}) \sum_{n=1}^{N_{\text{tot}}} \log P(s^n, a^n | \Lambda) \tag{10}$$

where $N_{\text{tot}}$ is the total number of the expert's demonstrated trajectories available in the dataset. As proven in [17], maximizing $L(\Lambda)$ with respect to $\Lambda$ is equivalent to minimizing an empirical approximation of the Kullback-Leibler divergence between the distributions of the agent's and expert's generated state-action trajectories. Note that when the final estimate $\widehat{\Lambda}$ produced by EM (Section 3) is plugged into (10), the resulting $P(s^n, a^n | \widehat{\Lambda})$ in (10) can be computed efficiently using a procedure similar to that in Section 3, as detailed in Appendix C. To avoid local maxima in gradient ascent, we initialize our EM algorithm with 20 random $\Lambda^0$ values and report the best result based on the $Q$ value of EM (Section 3).

To demonstrate the importance of modeling and learning stochastic transitions between locally consistent reward functions, the performance of our IRL algorithm is compared with that of its reduced variant assuming no change/switching of reward function within each trajectory, which is implemented by initializing $\tau_\omega(r_\theta, s, r_\theta) = 1$ for all $r_\theta \in \mathcal{R}$ and $s \in \mathcal{S}$ and deactivating the learning of $\tau_\omega$. In fact, it can be shown (Appendix D) that such a reduction, interestingly, is equivalent to EM clustering with MLIRL [2]. So, our IRL algorithm generalizes EM clustering with MLIRL, the latter of which has been empirically demonstrated in [2] to outperform many existing IRL algorithms, as discussed in Section 1.

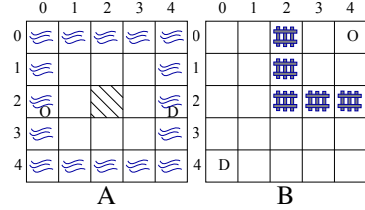

Figure 3: Grid worlds A (states $(0, 0)$, $(1, 1)$, and $(2, 2)$ are, respectively, examples of water, land, and obstacle), and B (state $(2, 2)$ is an example of barrier). 'O' and 'D' denote origin and destination.

**Simulated grid world A.** The environment (Fig. 3) is modeled as a $5 \times 5$ grid of states, each of which is either land, water, water and destination, or obstacle associated with the respective feature vectors (i.e., $\phi_s$) $(0, 1, 0)^\top$, $(1, 0, 0)^\top$, $(1, 0, 1)^\top$, and $(0, 0, 0)^\top$. The expert starts at origin $(0, 2)$ and any of its actions can achieve the desired state with $0.85$ probability. It has two possible reward functions, one of which prefers land to water and going to destination (i.e., $\theta = (0, 20, 30)^\top$), and the other of which prefers water to land and going to destination (i.e., $\theta' = (20, 0, 30)^\top$). The expert will only consider switching its reward function at states $(2, 0)$ and $(2, 4)$ from $r_{\theta'}$ to $r_\theta$ with $0.5$ probability and from $r_\theta$ to $r_{\theta'}$ with $0.7$ probability; its reward function remains unchanged at all other states. The feature measurements of $\varphi_s$ cannot be observed by the agent during the expert's demonstration. So, $\omega = \emptyset$ and $\tau_\omega$ is estimated using (8). We set $\gamma$ to $0.95$ and the number $|\mathcal{R}|$ of reward functions of the agent to 2.

Fig. 4a shows results of the average log-likelihood $L$ (10) achieved by our IRL algorithm, EM clustering with MLIRL, and the expert averaged over 4 random instances with varying number $N$ of expert's demonstrated trajectories. It can be observed that our IRL algorithm significantly outperforms EM clustering with MLIRL and achieves a $L$ performance close to that of the expert, especially when $N$ increases. This can be explained by its modeling of $\tau_\omega$ and its high fidelity in learning and predicting $\tau_\omega$: While our IRL algorithm allows switching of reward function within each trajectory, EM clustering with MLIRL does not.

We also observe that the accuracy of estimating the transition probabilities $\tau_\omega(r_\theta, s, .)$ $(\tau_\omega(r_{\theta'}, s, .))$ using (8) depends on the frequency and distribution of trajectories demonstrated by the expert with its reward function $R_{\theta_{t-1}^n} = r_\theta$ $(R_{\theta_{t-1}^n} = r_{\theta'})$ at time step $t-1$ and its state $s_t^n = s$ at time step $t$, which is expected. Those transition probabilities that are poorly estimated due to few relevant expert's demonstrated trajectories, however, do not hurt the $L$

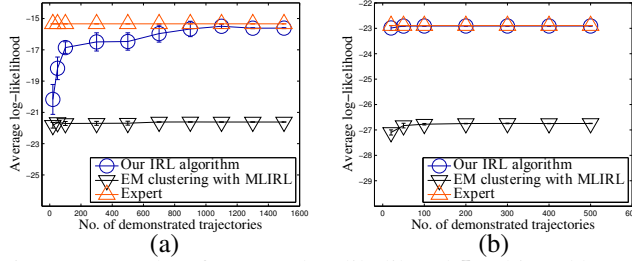

Figure 4: Graphs of average log-likelihood $L$ achieved by our IRL algorithm, EM clustering with MLIRL, and the expert vs. number $N$ of expert's demonstrated trajectories in simulated grid worlds (a) A ($N_{\text{tot}} = 1500$) and (b) B ($N_{\text{tot}} = 500$).

performance of our IRL algorithm by much because such trajectories tend to have very low probability of being demonstrated by the expert. In any case, this issue can be mitigated by using the generalized linear model (1) to represent $\tau_\omega$ and observing the feature measurements of $\varphi_s$ necessary for learning and computing $\tau_\omega$, as shown next.

**Simulated grid world B.** The environment (Fig. 3) is also modeled as a $5 \times 5$ grid of states, each of which is either the origin, destination, or land associated with the respective feature vectors (i.e. $\phi_s$) $(0,1)^\top$, $(1,0)^\top$, and $(0,0)^\top$. The expert starts at origin $(4,0)$ and any of its actions can achieve the desired state with $0.85$ probability. It has two possible reward functions, one of which prefers going to destination (i.e., $\theta = (30, 0)^\top$), and the other of which prefers returning to origin (i.e., $\theta' = (0, 30)^\top$). While moving to the destination, the expert will encounter barriers at some states with corresponding feature vectors $\varphi_s = (1,1)^\top$ and no barriers at all other states with $\varphi_s = (0,1)^\top$; the second component of $\varphi_s$ is used as an offset value in the generalized linear model (1). The expert's behavior of switching between reward functions is governed by a generalized linear model $\tau_\omega$ (1) with $r_{\widetilde{\theta}} = r_{\theta'}$ and transition weights $\omega_{r_\theta r_\theta} = (-11, 12)^\top$ and $\omega_{r_{\theta'} r_\theta} = (13, -12)^\top$. As a result, it will, for example, consider switching its reward function at states with barriers from $r_\theta$ to $r_{\theta'}$ with $0.269$ probability. We estimate $\tau_\omega$ using (9) and set $\gamma$ to $0.95$ and the number $|\mathcal{R}|$ of reward functions of the agent to $2$. To assess the fidelity of learning and predicting the stochastic transitions between reward functions at unvisited states, we intentionally remove all demonstrated trajectories that visit state $(2, 0)$ with a barrier.

Fig. 4b shows results of $L$ (10) performance achieved by our IRL algorithm, EM clustering with MLIRL, and the expert averaged over $4$ random instances with varying $N$. It can again be observed that our IRL algorithm outperforms EM clustering with MLIRL and achieves an $L$ performance comparable to that of the expert due to its modeling of $\tau_\omega$ and its high fidelity in learning and predicting $\tau_\omega$: While our IRL algorithm allows switching of reward function within each trajectory, EM clustering with MLIRL does not. Besides, the estimated transition function $\tau_{\widehat{\omega}}$ using (9) is very close to that of the expert, even at unvisited state $(2, 0)$. So, unlike using (8), the learning of $\tau_\omega$ with (9) can be generalized well across different states, thus allowing $\tau_\omega$ to be predicted accurately at any state. Hence, we will model $\tau_\omega$ with (1) and learn it using (9) in the next experiment.

**Real-world taxi trajectories.** The Comfort taxi company in Singapore has provided GPS traces of $59$ taxis with the same origin and destination that are map-matched [18] onto a network (i.e., comprising highway, arterials, slip roads, etc) of $193$ road segments (i.e., states). Each road segment/state is specified by a $7$-dimensional feature vector $\phi_s$: Each of the first six components of $\phi_s$ is an indicator describing whether it belongs to *Alexandra Road* (AR), *Ayer Rajah Expressway* (AYE), *Depot Road* (DR), *Henderson Road* (HR), *Jalan Bukit Merah* (JBM), or *Lower Delta Road* (LDR), while the last component of $\phi_s$ is the normalized shortest path distance from the road segment to destination. We assume that the $59$ map-matched trajectories are demonstrated by taxi drivers with a common set $\mathcal{R}$ of $2$ reward functions and the same transition function $\tau_\omega$ (1) for switching between reward functions, the latter of which is influenced by the normalized taxi speed constituting the first component of $2$-dimensional feature vector $\varphi_s$; the second component of $\varphi_s$ is used as an offset of value $1$ in the generalized linear model (1). The number $|\mathcal{R}|$ of reward functions is set to $2$ because when we experiment with $|\mathcal{R}| = 3$, two of the learned reward functions are similar. Every driver can deterministically move its taxi from its current road segment to the desired adjacent road segment.

Fig. 5a shows results of $L$ (10) performance achieved by our IRL algorithm and EM clustering with MLIRL averaged over 3 random instances with varying $N$. Our IRL algorithm outperforms EM clustering with MLIRL due to its modeling of $\tau_\omega$ and its high fidelity in learning and predicting $\tau_\omega$.

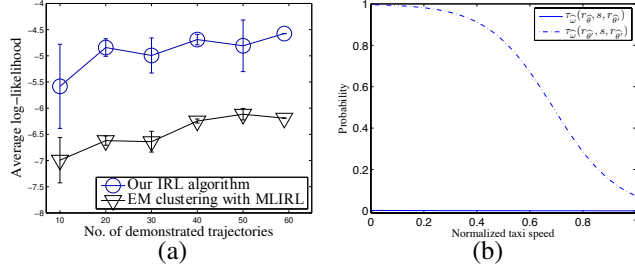

(a)       (b)

Figure 5: Graphs of (a) average log-likelihood $L$ achieved by our IRL algorithm and EM clustering with MLIRL vs. no. $N$ of taxi trajectories ($N_{\text{tot}} = 59$) and (b) transition probabilities of switching between reward functions vs. taxi speed.

To see this, our IRL algorithm is able to learn that a taxi driver is likely to switch between reward functions representing different intentions within its demonstrated trajectory: Reward function $r_{\widehat{\theta}}$ denotes his intention of driving directly to the destination (Fig. 6a) due to a huge penalty (i.e., reward weight -49) on being far from destination and a large reward (i.e., reward weight 35.7) for taking the shortest path from origin to destination, which is via JBM, while $r_{\widehat{\theta}'}$ denotes his intention of detouring to DR or JBM (Fig. 6b) due to large rewards for traveling on them (respectively, reward weights 30.5 and 23.7). As an example, Fig. 6c shows the most likely partition of a demonstrated trajectory into segments generated from locally consistent reward functions $r_{\widehat{\theta}}$ and $r_{\widehat{\theta}'}$, which is derived using our Viterbi algorithm (Section 3). It can be observed that the driver is initially in $r_{\widehat{\theta}'}$ on the slip road exiting AYE, switches from $r_{\widehat{\theta}'}$ to $r_{\widehat{\theta}}$ upon turning into AR to detour to DR, and remains in $r_{\widehat{\theta}}$ while driving along DR, HR, and JBM to destination. On the other hand, the reward functions learned by EM clustering with MLIRL are both associated with his intention of driving directly to destination (i.e., similar to $r_{\widehat{\theta}}$); it is not able to learn his intention of detouring to DR or JBM.

Fig. 5b shows the influence of normalized taxi speed (i.e., first component of $\varphi_s$) on the estimated transition function $\tau_{\widehat{\omega}}$ using (9). It can be observed that when the driver is in $r_{\widehat{\theta}}$ (i.e., driving directly to destination), he is very unlikely to change his intention regardless of taxi speed. But, when he is in $r_{\widehat{\theta}'}$ (i.e., detouring to DR or JBM), he is likely (unlikely) to remain in this intention if taxi speed is low (high). The demonstrated trajectory in Fig. 6c in fact supports this observation: The driver initially remains in $r_{\widehat{\theta}'}$ on the upslope slip road exiting AYE, which causes the low taxi speed. Upon turning into AR to detour to DR, he switches from $r_{\widehat{\theta}'}$ to $r_{\widehat{\theta}}$ because he can drive at relatively high speed on flat terrain.

## 5 Conclusion

This paper describes an EM-based IRL algorithm that can learn the multiple reward functions being locally consistent in different segments along a trajectory as well as the stochastic transitions between them. It generalizes EM-clustering with MLIRL and has been empirically demonstrated to outperform it on both synthetic and real-world datasets.

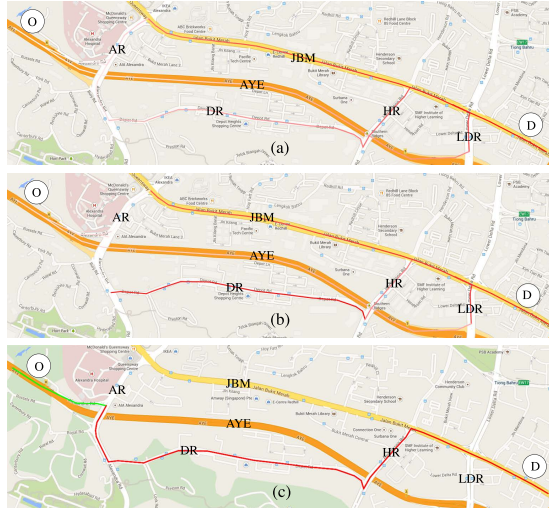

Figure 6: Reward (a) $r_{\widehat{\theta}}(s)$ and (b) $r_{\widehat{\theta}'}(s)$ for each road segment $s$ with $\widehat{\theta} = (7.4, 3.9, 16.3, 20.3, 35.7, 21.5, -49.0)^\top$ and $\widehat{\theta}' = (5.2, 9.2, 30.5, 15.0, 23.7, 21.5, -9.2)^\top$ such that more red road segments give higher rewards. (c) Most likely partition of a demonstrated trajectory from origin 'O' to destination 'D' into red and green segments generated by $r_{\widehat{\theta}}$ and $r_{\widehat{\theta}'}$, respectively.

For our future work, we plan to extend our IRL algorithm to cater to an unknown number of reward functions [6], nonlinear reward functions [12] modeled by Gaussian processes [4, 8, 13, 14, 15, 25], other dissimilarity measures described in Section 1, linearly-solvable MDPs [7], active learning with Gaussian processes [11], and interactions with self-interested agents [9, 10].

**Acknowledgments.** This work was partially supported by Singapore-MIT Alliance for Research and Technology Subaward Agreement No. 52 R-252-000-550-592.

## Footnotes

[1]If the state transition function is unknown, then it can be learned by optimizing the last term (5).

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
