[Supplementary Material · nips2015-localIRL-supplementary.pdf]

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

# A  Derivation of $Q$ Function of EM

The $Q$ function of EM in Section 3 can be reduced to the following sum of five terms:

$$
\begin{aligned}
&Q(\Lambda, \Lambda^i) \\
&= \sum_{r_{\theta^{1:N}}} P(r_{\theta^{1:N}}|s^{1:N}, a^{1:N}, \Lambda^i) \log P(r_{\theta^{1:N}}, s^{1:N}, a^{1:N}|\Lambda) \\
&= \sum_{r_{\theta^{1:N}}} \left( \prod_{n=1}^{N} P(r_{\theta^n}|s^n, a^n, \Lambda^i) \right) \times \sum_{n=1}^{N} \left( \log \nu(s_1^n) + \log \sigma(r_{\theta_0^n}) + \sum_{t=1}^{T_n - 1} \log t(s_t^n, a_t^n, s_{t+1}^n) \right. \\
&\hspace{6cm} \left. + \sum_{t=1}^{T_n} \left( \log \tau_\omega(r_{\theta_{t-1}^n}, s_t^n, r_{\theta_t^n}) + \log \pi_{\theta_t^n}(s_t^n, a_t^n) \right) \right) \\
&= \sum_{n=1}^{N} \log \nu(s_1^n) + \sum_{n=1}^{N} \sum_{r_\theta \in \mathcal{R}} P(R_{\theta_0^n} = r_\theta|s^n, a^n, \Lambda^i) \log \sigma(r_\theta) \\
&\quad + \sum_{n=1}^{N} \sum_{t=1}^{T_n} \sum_{r_\theta \in \mathcal{R}} P(R_{\theta_t^n} = r_\theta|s^n, a^n, \Lambda^i) \log \pi_\theta(s_t^n, a_t^n) \\
&\quad + \sum_{n=1}^{N} \sum_{t=1}^{T_n} \sum_{r_\theta, r_{\theta'} \in \mathcal{R}} P(R_{\theta_{t-1}^n} = r_\theta, R_{\theta_t^n} = r_{\theta'}|s^n, a^n, \Lambda^i) \times \log \tau_\omega(r_\theta, s_t^n, r_{\theta'}) \\
&\quad + \sum_{n=1}^{N} \sum_{t=1}^{T_n - 1} \log t(s_t^n, a_t^n, s_{t+1}^n) \\
&= \sum_{n=1}^{N} \log \nu(s_1^n) + \sum_{n=1}^{N} \sum_{r_\theta \in \mathcal{R}} P(R_{\theta_0^n} = r_\theta|s^n, a^n, \Lambda^i) \log \sigma(r_\theta) \\
&\quad + \sum_{n=1}^{N} \sum_{t=1}^{T_n} \sum_{r_\theta \in \mathcal{R}} P(R_{\theta_t^n} = r_\theta|s^n, a^n, \Lambda^i) \log \pi_\theta(s_t^n, a_t^n) \\
&\quad + \sum_{n=1}^{N} \sum_{t=1}^{T_n} \sum_{r_\theta, r_{\theta'} \in \mathcal{R}} P(R_{\theta_{t-1}^n} = r_\theta, s_t^n, R_{\theta_t^n} = r_{\theta'}|s^n, a^n, \Lambda^i) \times \log \tau_\omega(r_\theta, s_t^n, r_{\theta'}) \\
&\quad + \sum_{n=1}^{N} \sum_{t=1}^{T_n - 1} \log t(s_t^n, a_t^n, s_{t+1}^n) \,.
\end{aligned}
$$

# B  Efficient Algorithms for Computing $P(R_{\theta_t^n} = r_\theta|s^n, a^n, \Lambda^i)$ and $P(R_{\theta_{t-1}^n} = r_\theta, S_t^n = s, R_{\theta_t^n} = r_{\bar{\theta}}|s^n, a^n, \Lambda^i)$

We will first describe an efficient procedure inspired by the Baum-Welch algorithm [3] to compute $P(R_{\theta_0^n} = r_\theta|s^n, a^n, \Lambda^i)$, the output of which is used in (6) and (7).

**Forward variables** $\alpha_{n,t,r_{\theta^i}}$ for all $n = 1, \ldots, N$, $t = 1, \ldots, T_n$, and $r_{\theta^i} \in \mathcal{R}$ are computed recursively as follows:

$$
\begin{aligned}
\alpha_{n,0,r_{\theta^i}} &\triangleq P(R_{\theta_0^n} = r_\theta|\Lambda^i) = \sigma^i(r_{\theta^i}) \,, \\
\alpha_{n,1,r_{\theta^i}} &\triangleq P(s_1^n, a_1^n, R_{\theta_1^n} = r_\theta|\Lambda^i) \\
&= \sum_{r_{\bar{\theta}^i} \in \mathcal{R}} \alpha_{n,0,r_{\bar{\theta}^i}} \times \widehat{\nu}(s_1^n) \times \pi_{\bar{\theta}^i}(s_1^n, a_1^n) \times \tau_{\omega^i}(r_{\bar{\theta}^i}, s_1^n, r_{\theta^i}) \,, \\
\alpha_{n,t,r_{\theta^i}} &\triangleq P(s_1^n, a_1^n, \ldots, s_t^n, a_t^n, R_{\theta_t^n} = r_\theta|\Lambda^i) \\
&= \sum_{r_{\bar{\theta}^i} \in \mathcal{R}} \alpha_{n,t-1,r_{\bar{\theta}^i}} \times t(s_{t-1}^n, a_{t-1}^n, s_t^n) \times \pi_{\bar{\theta}^i}(s_t^n, a_t^n) \times \tau_{\omega^i}(r_{\bar{\theta}^i}, s_t^n, r_{\theta^i}) \,;
\end{aligned}
$$

where $\widehat{\nu}$, $\sigma^i$, $\theta^i$, $\bar{\theta}^i$, and $\tau_{\omega^i}$ denote estimates for $\nu$, $\sigma$, $\theta$, $\bar{\theta}$, and $\tau_\omega$ at iteration $i$, respectively (Section 3).

**Backward variables** $\beta_{n,t,r_{\theta i}}$ for all $n = 1, \ldots, N$, $t = 1, \ldots, T_n$, and $r_{\theta i} \in \mathcal{R}$ are computed recursively as follows:

$$\beta_{n,T_n,r_{\theta i}} \triangleq 1 ,$$

$$\begin{aligned}
\beta_{n,t,r_{\theta i}} &\triangleq P(s_{t+1}^n, a_{t+1}^n, \ldots, s_{T_n}^n, a_{T_n}^n | s_t^n, a_t^n, R_{\theta_t^n} = r_\theta, \Lambda^i) \\
&= \sum_{r_{\bar{\theta} i} \in \mathcal{R}} \beta_{n,t+1,r_{\bar{\theta} i}} \times t(s_t^n, a_t^n, s_{t+1}^n) \times \pi_{\bar{\theta} i}(s_{t+1}^n, a_{t+1}^n) \times \tau_{\omega^i}(r_\theta, s_{t+1}^n, r_{\bar{\theta}}) ,
\end{aligned}$$

$$\begin{aligned}
\beta_{n,0,r_{\theta i}} &\triangleq P(s_1^n, a_1^n, \ldots, s_{T_n}^n, a_{T_n}^n | R_{\theta_0^n} = r_\theta, \Lambda^i) \\
&= \sum_{r_{\bar{\theta} i} \in \mathcal{R}} \beta_{n,1,r_{\bar{\theta} i}} \times \widehat{\nu}(s_1^n) \times \pi_{\bar{\theta} i}(s_1^n, a_1^n) \times \tau_{\omega^i}(r_{\theta i}, s_1^n, r_{\bar{\theta} i}) .
\end{aligned}$$

Then, for all $n = 1, \ldots, N$, $t = 0, \ldots, T_n$, and $r_\theta \in \mathcal{R}$,

$$P(R_{\theta_t^n} = r_\theta | s^n, a^n, \Lambda^i) = \frac{\alpha_{n,t,r_{\theta i}} \times \beta_{n,t,r_{\theta i}}}{P(s^n, a^n | \Lambda^i)}$$

where $P(s^n, a^n | \Lambda^i) = \sum_{r_{\bar{\theta} i} \in \mathcal{R}} \alpha_{n,0,r_{\bar{\theta} i}} \times \beta_{n,0,r_{\bar{\theta} i}}$.

Using the above forward and backward variables, we will next describe an efficient procedure to compute $P(R_{\theta_{t-1}^n} = r_\theta, s, R_{\theta_t^n} = r_{\bar{\theta}} | s^n, a^n, \Lambda^i)$, the output of which is needed in (8) and (9).

**Transition variables** $\gamma_{n,t,r_{\theta i},s,r_{\bar{\theta} i}}$ for all $n = 1, \ldots, N$, $t = 1, \ldots, T_n$, $r_{\theta i}, r_{\bar{\theta} i} \in \mathcal{R}$, and $s \in \mathcal{S}$ are computed recursively as follows:

$$\begin{aligned}
\gamma_{n,1,r_{\theta i},s,r_{\bar{\theta} i}} &\triangleq P(R_{\theta_0^n} = r_\theta, S_1^n = s, R_{\theta_1^n} = r_{\bar{\theta}} | s^n, a^n, \Lambda^i) \\
&= \frac{I_1^n \times \alpha_{n,0,r_{\theta i}} \times \widehat{\nu}(s) \times \pi_{\bar{\theta} i}(s, a_1^n) \times \tau_{\omega^i}(r_{\theta i}, s, r_{\bar{\theta} i}) \times \beta_{n,1,r_{\bar{\theta} i}}}{P(s^n, a^n | \Lambda^i)} ,
\end{aligned}$$

$$\begin{aligned}
\gamma_{n,t,r_{\theta i},s,r_{\bar{\theta} i}} &\triangleq P(R_{\theta_{t-1}^n} = r_\theta, S_t^n = s, R_{\theta_t^n} = r_{\bar{\theta}} | s^n, a^n, \Lambda^i) \\
&= \frac{I_t^n \times \alpha_{n,t-1,r_{\theta i}} \times t(s_{t-1}^n, a_{t-1}^n, s) \times \pi_{\bar{\theta} i}(s, a_t^n) \times \tau_{\omega^i}(r_{\theta i}, s, r_{\bar{\theta} i}) \times \beta_{n,t,r_{\bar{\theta} i}}}{P(s^n, a^n | \Lambda^i)} ;
\end{aligned}$$

where $I_t^n$ is an indicator variable of value 1 if $s_t^n = s$, and 0 otherwise.

## C  Efficient Algorithm for Computing Likelihood $P(s^n, a^n | \widehat{\Lambda})$ of a Trajectory

Given the final estimate $\widehat{\Lambda} = (\widehat{\nu}, \widehat{\sigma}, \{\widehat{\theta} | r_{\widehat{\theta}} \in \mathcal{R}\}, \tau_{\widehat{\omega}})$ for the unknown parameters $\Lambda$ produced by EM (Section 3), $P(s^n, a^n | \widehat{\Lambda})$ can be evaluated by recursively computing $P(R_{\theta_T^n} = r_\theta, (s_t^n)_{t=1}^T, (a_t^n)_{t=1}^T | \widehat{\Lambda})$ for $T = 1, \ldots, T_n$ as follows:

$$P(s^n, a^n | \widehat{\Lambda}) = \sum_{r_\theta \in \mathcal{R}} P(R_{\theta_{T_n}^n} = r_\theta, s^n, a^n | \widehat{\Lambda}) ,$$

$$\begin{aligned}
&P(R_{\theta_T^n} = r_\theta, (s_t^n)_{t=1}^T, (a_t^n)_{t=1}^T | \widehat{\Lambda}) \\
&= \sum_{r_{\theta'} \in \mathcal{R}} P(R_{\theta_{T-1}^n} = r_{\theta'}, R_{\theta_T^n} = r_\theta, (s_t^n)_{t=1}^T, (a_t^n)_{t=1}^T | \widehat{\Lambda}) \\
&= t(s_{T-1}^n, a_{T-1}^n, s_T^n) \, \pi_{\widehat{\theta}}(s_T^n, a_T^n) \times \sum_{r_{\theta'} \in \mathcal{R}} P(R_{\theta_{T-1}^n} = r_{\theta'}, (s_t^n)_{t=1}^{T-1}, (a_t^n)_{t=1}^{T-1} | \widehat{\Lambda}) \, \tau_{\widehat{\omega}}(r_{\widehat{\theta}'}, s_T^n, r_{\widehat{\theta}}) ,
\end{aligned}$$

$$\begin{aligned}
&P(R_{\theta_1^n} = r_\theta, s_1^n, a_1^n | \widehat{\Lambda}) \\
&= \sum_{r_{\theta'} \in \mathcal{R}} P(R_{\theta_0^n} = r_{\theta'}, R_{\theta_1^n} = r_\theta, s_1^n, a_1^n | \widehat{\Lambda}) \\
&= \widehat{\nu}(s_1^n) \, \pi_{\widehat{\theta}}(s_1^n, a_1^n) \sum_{r_{\theta'} \in \mathcal{R}} \widehat{\sigma}(r_{\widehat{\theta}'}) \, \tau_{\widehat{\omega}}(r_{\widehat{\theta}'}, s_1^n, r_{\widehat{\theta}}) .
\end{aligned}$$

## D   Reduction to EM Clustering with MLIRL

If we assume no change/switching of reward function within each trajectory (in other words, each trajectory is assumed to be generated by a single reward function) by initializing $\tau_\omega(r_\theta, s, r_\theta) = 1$ for all $r_\theta \in \mathcal{R}$ and $s \in \mathcal{S}$ and deactivating the learning of $\tau_\omega$ (Section 4), then

$$
P(R_{\theta_t^n} = r_\theta | s^n, a^n, \Lambda^i)
$$
$$
= \frac{\alpha_{n,t,r_{\theta^i}} \times \beta_{n,t,r_{\theta^i}}}{P(s^n, a^n | \Lambda^i)}
$$
$$
= \frac{\sigma^i(r_{\theta^i}) \widehat{\nu}(s_1^n) \prod_{t'=1}^{T_n} \pi_{\theta^i}(s_{t'}^n, a_{t'}^n) \prod_{t'=1}^{T_n - 1} t(s_{t'}^n, a_{t'}^n, s_{t'+1}^n)}{P(s^n, a^n | \Lambda^i)} \, ,
$$

for all $n = 1, \dots, N$, $t = 0, \dots, T_n$, and $r_\theta \in \mathcal{R}$, which implies that the value of $P(R_{\theta_t^n} = r_\theta | s^n, a^n, \Lambda^i)$ remains the same for $t = 0, \dots, T_n$. Then, $P(R_{\theta_t^n} = r_\theta | s^n, a^n, \Lambda^i)$ can be simplified to $P(r_\theta | s^n, a^n, \Lambda^i)$ without ambiguity. As a result, the $Q$ function of EM in Section 3 can be reduced to

$$
Q(\Lambda, \Lambda^i)
$$
$$
= \sum_{n=1}^N \log \nu(s_1^n) + \sum_{n=1}^N \sum_{r_\theta \in \mathcal{R}} P(r_\theta | s^n, a^n, \Lambda^i) \log \sigma(r_\theta)
$$
$$
+ \sum_{n=1}^N \sum_{r_\theta \in \mathcal{R}} P(r_\theta | s^n, a^n, \Lambda^i) \sum_{t=1}^{T_n} \log \pi_\theta(s_t^n, a_t^n)
$$
$$
+ \sum_{n=1}^N \sum_{t=1}^{T_n - 1} \log t(s_t^n, a_t^n, s_{t+1}^n)
$$
$$
= \sum_{n=1}^N \sum_{r_\theta \in \mathcal{R}} P(r_\theta | s^n, a^n, \Lambda^i) \log \sigma(r_\theta)
$$
$$
+ \sum_{n=1}^N \sum_{r_\theta \in \mathcal{R}} P(r_\theta | s^n, a^n, \Lambda^i) \times \log \left( \nu(s_1^n) \prod_{t=1}^{T_n} \pi_\theta(s_t^n, a_t^n) \prod_{t=1}^{T_n - 1} t(s_t^n, a_t^n, s_{t+1}^n) \right)
$$
$$
= \sum_{n=1}^N \sum_{r_\theta \in \mathcal{R}} P(r_\theta | s^n, a^n, \Lambda^i) \log \sigma(r_\theta) + \sum_{n=1}^N \sum_{r_\theta \in \mathcal{R}} P(r_\theta | s^n, a^n, \Lambda^i) \log P(s^n, a^n | r_\theta)
$$

where

$$
P(s^n, a^n | r_\theta) = \nu(s_1^n) \prod_{t=1}^{T_n} \pi_\theta(s_t^n, a_t^n) \prod_{t=1}^{T_n - 1} t(s_t^n, a_t^n, s_{t+1}^n)
$$

is the probability of the expert's $n$-th demonstrated state-action trajectory being generated by the reward function $r_\theta$. The expression in the last equality of the above $Q$ function is exactly the $Q$ function of EM clustering with MLIRL [2].

The likelihood $P(s^n, a^n | \widehat{\Lambda})$ of a trajectory (Appendix C) needed for computing (10) can also be reduced to

$$
P(s^n, a^n | \widehat{\Lambda})
$$
$$
= \sum_{r_\theta \in \mathcal{R}} P(r_\theta, s^n, a^n | \widehat{\Lambda})
$$
$$
= \sum_{r_\theta \in \mathcal{R}} P(r_\theta | \widehat{\Lambda}) P(s^n, a^n | r_\theta, \widehat{\Lambda})
$$
$$
= \sum_{r_\theta \in \mathcal{R}} \widehat{\sigma}(r_{\widehat{\theta}}) \, \widehat{\nu}(s_1^n) \prod_{t=1}^{T_n} \pi_{\widehat{\theta}}(s_t^n, a_t^n) \prod_{t=1}^{T_n - 1} t(s_t^n, a_t^n, s_{t+1}^n) \, .
$$