[Reviews · NeurIPS 2015]

Submitted by Assigned_Reviewer_1

The problem formulation (Sec.2) was new to me, though the proposed approach (Sec.3) draws on some standard methodology for inference in graphical models (i.e. EM, Viterbi).

For such a conventional algorithm to have impact, I would expect to see a compelling application of the new model.

The empirical analysis presents results for two small simulated grid words, which confirm good performance, but don't provide compelling evidence that the new model is justified.

Some of the challenges of EM (e.g. convergence to local optimum, initialization, picking the number of reward functions) are not discussed - did this never occur in the experiments?

The third experiment uses real-world taxi trajectories, yet I don't quite understand why it's better to infer two separate reward functions, rather than incorporate the speed (which is the main feature in the reward transition function) directly in the reward parameterization (theta).

Following the author response: The response did clarify one misconception I had about the setting for the 3rd experiment (regarding whether the taxi speed should be incorporated as a reward variable). This improved my appreciation for the paper's motivation and setting.
Summary: The paper tackles the problem of IRL from multiple reward functions.

I had some doubts about the problem setting, though the discussion and other reviews raised some interesting points in support of this.

The algorithmic approach is well-motivated, and clearly explained, however I did not see anything new in this aspect.

Submitted by Assigned_Reviewer_2

This paper addresses the problem of inverse reinforcement learning when the agent can change it's objective during the recording of trajectories. This results in a transition between several reward functions that explain only locally the trajectory of the observed agent. Transition probabilities between reward functions are unknown. The author propose a cascade of an EM and Viterbi algorithms to discover the reward functions and the segments on which they are valid.

The paper is quite well written. Yet the state of the art about IRL stops in 2012. There has been quite a lot of work since then and I'm a bit surprised that the authors didn't think it was worth mentioning recent works (especially those that do not require to solve MDPs iteratively).

The authors propose a very complex model to address the problem at sight and I'm not sure it is worth to make this effort. Indeed, if I understood correctly, the number of reward functions has to be known in advance to run the EM algorithm. Therefore, I don't see why this problem cannot be modelled by adding a dimension to the state space for the index of the current reward function. As the reward function is not observable, this is a hidden variable which makes the problem to be modelled as a POMDP. But there are methods for IRL in POMDPs.

Also, the use of the Viterbi algorithm implies that the transition probabilities between states are known. This constrains even more the setting and makes it usable in very few situations.

Finally, this method can only be used for analysing the behaviour of an agent but not really to find an optimal control for the environment. To me, this actually means that a special care has to be taken to ensure that the learnt reward functions will be interpretable by a human. This is not mentioned in the paper but it is a quite important issue. When the state space is multidimensional, it is very hard to represent the reward function unless it is very sparse for instance. So how would the authors add constraints in this model to be sure to enable the analysis of the behaviour from the learnt reward?
Summary: Globally, I think the proposed model is too complex with respect to the targeted problem and the algorithms used are too restrictive to make this model applicable to real world problems.

Submitted by Assigned_Reviewer_3

The authors propose an IRL method that learns locally consistent reward functions from expert demonstrations. To do so, they suppose the existence of a transition function for switching between reward functions which depends on the rewards and the current state and allows them to build a graphical model of their problem.

Their algorithm consists in maximizing the log-likelihood of the expert's demonstrated trajectories depending on some parameters which are the original distributions of states and rewards, the local rewards and the transition function between rewards. To do so, they use the expectation-maximisation (EM) method. Then, via the Viterbi algorithm, they are able to partition the trajectories into segments with local consistent rewards.

Strengths of the paper:

1. The authors leverage existing and classical methods from the machine learning and optimization fields such as EM, Viterbi, Value iteration and gradient ascent in order to build their algorithm. This will allow the community to easily reproduce their results. 2. The experiments are conducted on synthetic and real-world data. They compare their method to MLIRL which does not use locally consistent rewards and which is the canonical choice to compare to as their algorithm is a generalization of MLIRL. The results presented show the superiority of their method over MLIRL. 3. The idea presented by the authors is original as far as I know.

Weaknesses of the paper:

1. The paper is very dense ( the figures are incorporated in the text) which makes the reading difficult. 2. The algorithm proposed needs the knowledge of the dynamics and the number of rewards. The authors, as future works, plan to extend their algorithm to unknown number of rewards, however they do not mention to get rid off the knowledge of the dynamics. Could the authors comment on that as some IRL algorithms do not need a perfect knowledge of the dynamics?

3. The method needs to solve iteratively MDPs when learning the reward functions. For each theta in the gradient ascent a MDP needs to be solved. Is this prohibitive for huge MDPs? Is there a way to avoid that step? The action-value function Q is defined via a softmax operator in order to have a derivable policy, does it allow to solve more efficiently the MDP? 4. The authors are using gradient ascent in the EM method, could they comment on the concavity of their criteria? 5. In the experiments (gridworlds), the number of features for the states is very small and thus it is understandable that a reward which is linear on the features will perform badly. Do the authors consider comparing their method to an IRL method where the number of features defining the states is greater? This is the main problem that I have with the experiments, the features used are not expressive enough to consider using a classical IRL method and this can explain why MLIRL performs badly and that its performance does not improve when the number of expert trajectories grows. 6. The performance is measured by the average log-likelihood of the expert's demonstrated trajectories which is the criterion maximized by the algorithm. I think that a more pertinent measure would be the value function of the policy produced by the optimization of the reward obtained by the algorithm. Could the authors comment on that and explain why their performance metric is more appropriate?

Summary: This paper presents a new idea consisting in learning locally consistent rewards in order to solve the IRL problem. The method is original and relies on classical machine learning methods. However, I have some concerns on the efficiency of the method for large MDPs and when the dynamics is not provided.

Summary: This paper presents a new idea consisting in learning locally consistent rewards in order to solve the IRL problem. The method is original and relies on classical machine learning methods. However, I have some concerns on the efficiency of the method for large MDPs and when the dynamics is not provided.

Submitted by Assigned_Reviewer_4

The paper proposes not assuming that even with IRL that there is one reward function but different parts of the state space.

in terms of the empirical results I wonder what would happen with fig 5 a with infinite data as well as the the impact on extracting optimal policies. So wouldn't the MLIRL with multiple intentions with enough data and enough underlying rewards would produce similar results?

How worse off, computationally, will the algorithm be against one that just assumes a single reward per trajectory if then you choose a # of reward number greater than 1 but actually there was only one reward?

How would one also protect against overfitting in this case, especially with the increase in parameters?
Summary: An interesting approach to IRL, further breaking possible paths as having different motivations (rewards).

Author Feedback
Author rebuttal: We thank all reviewers for their feedback, which will be considered when revising our paper.

Reviewer1

We explained in lines 92-95,142-144 that by representing the index of reward function as latent state component in POMDP, its observation model cannot be easily specified nor learned from expert's trajectories of states, actions & varphi_s, which invalidates the use of IRL for POMDP[4].

The state transition probabilities are assumed to be known as we want to focus on learning the reward functions & their transitions, which is the core of IRL.
Our approach can be easily extended to learn the unknown state transitions by modeling with a logistic model and optimizing the last term (5) in EM's Q function.
Another option is to learn the state transitions separately by counting the frequency of next state given the current state and action using the (state, action, next state) tuples from experts' trajectories.

Besides analyzing agent's behavior, our algorithm can find optimal control. For example, in taxi experiment, we can use the learned model to produce preferred route choices to guide drivers. To enable behavior analysis, the Viterbi algorithm identifies states where experts are likely to switch between reward functions; such states can be investigated for the resulting causes (lines 99-103). Regarding the reward function, similar to existing IRL algorithms, we assume linear combination of state features (line 114) & the weights represent preference over features.

Reviewer2

2. Our approach can be easily extended to learn the state transition dynamics by modeling with a logistic model and optimizing the last term (5) in EM's Q function.
Another option is to learn the state transitions separately by counting the frequency of next state given the current state and action using the (state, action, next state) tuples from experts' trajectories.

3. Our EM can use an MDP solution if it yields a stochastic policy differentiable wrt theta. Softmax is one such option that we used. Another option is the linearly solvable MDP (LMDP) that is more efficient than standard iterative MDP solvers. This class includes max entropy IRL, which is empirically shown in [2] to be outperformed by MLIRL, hence explaining our choice. We observe that our EM can use LMDP as its policy is stochastic and differentiable.

4. It is possible for EM to reach local maxima. So, for each experiment, we run 15 initializations and choose the best result based on EM's Q function.

5. We agree that in some cases, by increasing the number of features, a single reward function is enough to explain the expert's behavior. But, we focus on problems where no single reward function can explain the entire expert's trajectory, which motivates the need of multiple locally consistent reward functions (lines 76-78). Experiments are tailored to address this.

6. If we understand your proposed performance measure correctly, it requires the true reward functions, which are often not available in real-world datasets. In contrast, our metric directly uses the available demonstrated trajectories only.

Reviewer3

MLIRL with multiple intentions and enough data and reward functions cannot produce same results as our algorithm if the expert switches between reward functions within a demonstrated trajectory. This is because it does not allow reward function to switch within a trajectory, while ours does (lines 295-304,322-323). But, if the expert does not change his intention within a trajectory, MLIRL with multiple intentions is equivalent to our algorithm.

Supposing we compare with MLIRL using one reward function, our algorithm with multiple reward functions incurs most time in solving MDPs. So, its incurred time increases linearly in number of reward functions.

Model complexity is determined by number of reward functions, which can be controlled to reduce overfitting. A better solution (future work) is to use a Bayesian nonparametric approach to learn the number of reward functions from data (line 429).

Reviewer4

We faced the local optimum issue. So, we initialize 15 times for each experiment and choose best result based on EM Q-function. In taxi experiment, we tried 3 reward functions initially, but 2 are similar. Thus, we reduce to 2. A better solution (future work) is to use a Bayesian nonparametric approach to learn the number of reward functions from data (line 429).

Expert driver can only observe its taxi speed while traveling along that road. Without prior taxi speed information over all states, the expert cannot exploit it within MDP planning. So, we use taxi speed as the feature affecting reward function switching during execution, but not the feature constituting the reward functions (see Remark 1).

Reviewer5
The gradient of MDP Q function is computed based on a recursive equation by assuming softmax operation for its policy. Yes, we have to compute optimal policy for each reward function.